# Hospital at Home Following Allogeneic Hematopoietic Stem Cell Transplantation: An Economic Analysis

**DOI:** 10.3390/healthcare13141648

**Published:** 2025-07-08

**Authors:** Vinod Mishra, Tobias Gedde-Dahl, Mats Remberger, Grethe Solvang, Kristin Lien Selvaag, Arne Fosseng, Ingerid W. Abrahamsen, Anders E. Myhre, Terje P. Hagen, Geir E. Tjønnfjord

**Affiliations:** 1Department of Governance, Oslo University Hospital, P.O. Box 4950 Nydalen, 0424 Oslo, Norway; 2Department of Health Management and Health Economics, Faculty of Medicine, University of Oslo, P.O. Box 1130 Blindern, 0318 Oslo, Norway; t.p.hagen@medisin.uio.no; 3Department of Hematology, Oslo University Hospital, P.O. Box 4950 Nydalen, 0424 Oslo, Norway; tgeddeda@ous-hf.no (T.G.-D.); mats.remberger@akademiska.se (M.R.); uxgrlv@ous-hf.no (G.S.); krselv@ous-hf.no (K.L.S.); inabra@ous-hf.no (I.W.A.); andmyhr@ous-hf.no (A.E.M.); gtjonnfj@ous-hf.no (G.E.T.); 4Institute of Clinical Medicine, University of Oslo, P.O. Box 1171 Blindern, 0318 Oslo, Norway; 5Division of Cancer Medicine, Oslo University Hospital, P.O. Box 4950 Nydalen, 0424 Oslo, Norway; arnef@ous-hf.no

**Keywords:** advanced home care, hospital at home, hospital costs, Norway

## Abstract

Background: Advanced home care is becoming increasingly common for cancer patients and serves as a viable alternative to inpatient hospital care. The transition to home care is driven by both the rising costs of healthcare and evidence indicating better quality of care. This study aims to compare the costs of hospital-at-home treatment and in-hospital care for patients undergoing allo-HSCT. Methods: The cost analysis was conducted as a case–control study comparing the costs of allo-HSCT at home (HaH) to the costs of allo-HSCT for patients receiving in-hospital care (INH). The cost evaluation was conducted from the hospital’s perspective, which means that costs incurred outside the hospital setting were not included. Post-procedural costs for the first year after allo-HSCT included all readmissions and outpatient visits at Oslo University Hospital. Results: The cost for the peritransplant period could be reduced by up to 33% by treating allo-HSCT recipients at home instead of in the hospital. During the study period, 24% of the allo-HSCT recipients were treated at home, but our data from 2021 and 2022 indicate that at least a third of the patients scheduled for allo-HSCT are candidates for HaH. Conclusions: The findings demonstrate that patients in advanced home care experience significantly lower total costs compared to those receiving in-hospital treatment.

## 1. Introduction

Allogeneic hematopoietic stem cell transplantation (allo-HSCT) is a well-established curative treatment for malignant and non-malignant hematological diseases [1,2]. Traditionally, patients undergoing allo-HSCT have been primarily cared for as inpatients, often in isolation, at most transplant centers. However, a pioneering study by Svahn and Bjurman [3] introduced the concept of treating patients at home following the infusion of stem cells. This approach was well-received by patients [4], who reported increased physical activity and reduced reliance on parenteral nutrition, as well as other supportive measures like antibiotics and analgesics [5,6,7]. Additionally, hospital at home (HaH) for patients undergoing allo-HSCT was found to be safe, beneficial in terms of outcome, and more cost-effective [8,9,10]. These results have been confirmed by subsequent studies [11,12,13,14,15].

In 2018, Oslo University Hospital launched a pilot home care program for allo-HSCT recipients, which subsequently became a standard treatment option in 2019. Consistent with the findings of Svahn and Bjurman [3], our study demonstrates that home care is not only safe, but also highly appreciated by patients and their family caregivers [14].

The purpose of this study was to compare the costs of HaH and in-hospital treatment (INH) for allo-HSCT patients.

## 2. Materials and Methods

### 2.1. Design

This is a retrospective case–control study. The data collected for this study consists of economic information that does not require ethical approval.

### 2.2. Sample

Since January 2019, patients admitted for allo-HSCT and residing within a 45–60 min driving distance from Oslo University Hospital have been given the option of HaH if deemed suitable by the transplantation team. To qualify for HaH, certain requirements must be met:≥18 years of age.Home care approved as medically suitable by the transplant team.An informal caregiver (family member or friend) available at all times.The patient and caregiver must be capable of understanding and following instructions, as well as communicating proficiently in Norwegian or English.The home must meet specific hygienic standards.

As of May 2020, the hospital has procured four 2-bedroom apartments to accommodate all eligible patients admitted for allo-HSCT. From October 2018 until the conclusion of our observation period in September 2023, 143 out of a total of 586 allo-HSCT patients were approved for HaH.

### 2.3. Hospital Care

In-hospital treatment for HaH patients encompassed the conditioning phase and the infusion of stem cells. Throughout this period, eligible patients and their informal caregivers received comprehensive instructions regarding the procedures during the home care phase and how to engage with hospital staff if needed. They were furnished with information on when and where to seek assistance, dietary guidelines for the patient, and recommendations regarding outdoor activities. Overall, patients were encouraged to engage in outdoor activities. Visitors were permitted at the HaH units, but it was imperative that they were free from any infections.

### 2.4. Home Care

The transfer of patients to HaH occurred no earlier than the day following the stem cell infusion. Patients receiving home care were provided with antibacterial, antiviral, and antifungal prophylaxis according to institutional guidelines. During the initial post-transplantation phase, a skilled nurse visited the patient once daily from Monday to Friday. These visits involved monitoring vital signs, conducting patient examinations, collecting blood samples, and administering medications and transfusions.

The attending physician would contact the patient every afternoon to confirm the treatment plan or make any necessary adjustments. On weekends, patients would visit the transplant center and be examined by a nurse, and if needed, by a physician, before returning to the home care unit. If deemed necessary, such as in cases of new onset of infections, difficulty maintaining oral intake, or at the request of the patient, informal caregiver, or medical staff, the patient would be readmitted to the hospital. If the patient remained stable and felt well after observation in the ward, home care could continue with ceftriaxone administered once daily in the event of an infection. Subsequent antibacterial therapy was adjusted based on the findings from blood cultures. Additional blood cultures were obtained when patients experienced high fever and/or chills.

### 2.5. Cost Analysis

The cost analysis was conducted as a case–control study comparing the costs of allo-HSCT at home (HaH) to the costs of allo-HSCT for patients receiving in-hospital care (INH). All costs were adjusted to 2023 prices using the consumer price index and converted to US dollars (USD) at an exchange rate of USD 1 to NOK 10.8. The length of stay (LOS) was calculated by subtracting the admission date from the discharge date for both hospital and HaH stays.

For the INH patient group, we gathered total costs and costs for different cost categories, including the use of ICU and laboratories for DRG 481B, from the hospital’s “cost per patient” system for the fiscal year 2019. Initial cost calculations involved two data sets: one for patient-related or direct costs and another for non-patient-related or indirect costs [16]. These costs were summarized as the total cost per patient for each defined time period. Patient-level costs were based on clinical pathways identified through the hospital’s patient administrative systems. First, unit costs for healthcare personnel (e.g., nursing hours), procedures, and items were calculated for each clinical pathway. Unit costs included an estimated overhead cost of 15% of the total cost. Second, the unit cost was multiplied by the patient’s length of hospital stay measured in days or the number of items used at different stages of the clinical pathway, and then summed to obtain the total patient cost [17].

The cost evaluation was conducted from the hospital’s perspective, which means that costs incurred outside the hospital setting, such as at other hospitals or primary care facilities, and the loss of normal working days and wages for patients and their informal caregivers, were not included. Post-procedural costs for the first year after allo-HSCT included all readmissions and outpatient visits at Oslo University Hospital.

During the hospital stay, the costs for the conditioning and transplantation phases were similar for both groups, while we collected separate data for ward costs during the home care phase for HaH patients. In the “base case” calculations, we considered the hospital ward costs as fixed per day for the INH patients and calculated cost savings as the difference between the total hospital ward costs and total HaH ward costs.

The hospital ward costs included personnel costs for nurses and physicians, among others, while the HaH ward costs additionally included costs related to the transportation of healthcare personnel and house leasing.

The assumption of fixed ward costs across different phases of the care pathway reflects local staffing guidelines at our hospital. In sensitivity analyses, we therefore assumed that ward costs differed across the different phases for patients undergoing hospital treatment. In alternative 1, we assumed that the daily ward costs during the post-transplantation phase were 10% lower, and in alternative 2, 20% lower than in the base case (with the ward costs of the in-hospital phases correspondingly higher).

## 3. Results

### 3.1. Patients

Between January 2019 and September 2023, a total of 586 patients were included in the study, with 143 of them receiving treatment at HaH (Table 1). Three patients who were initially accepted for HaH were deemed unfit for home care on day +1 by the attending physician and were subsequently transferred to in-hospital care for the remainder of their stay. These patients had all undergone reduced-intensity conditioning (RIC). A few patients were unable to be offered HaH due to restrictions on the supply side, specifically the unavailability of apartments. One pilot patient underwent transplantation in October 2018, while the remaining patients were transplanted between 2019 and 2022. The proportion of patients receiving HaH treatment increased from 2019 to 2022, followed by a preliminary reduction in 2023. On average, 24.4% of the patients were treated at HaH during the study period.

There was a significant difference in median age between the two groups, with the HaH patients having a median age of 54 years (range 17–74) compared to 59 years (range 15–75) for the hospital group (Table 2). Among the HaH patients, 86 patients (60%) were male and 57 patients (40%) were female, which is a comparable distribution to that of the patients receiving hospital care. The proportion of patients receiving myeloablative conditioning (MAC) and total body irradiation as part of the conditioning was higher among the HaH patients (*p* < 0.005). Additionally, GvHD prophylaxis with cyclosporine and methotrexate was more frequently used for patients in HaH. Table 3.

No patients died while receiving HaH treatment, and no serious emergency situations occurred. One patient developed septicemia while at the HaH unit but was promptly admitted to the hospital within 30 min and successfully treated. For additional patient characteristics, see Table 2.

### 3.2. Cost Analyses

The average cost per patient for those receiving in-hospital (INH) care was USD 104,530, with hospital ward costs accounting for USD 86,394 or 82.7% of the total cost (Table 4). Other major cost drivers included ICU costs (USD 6386) and laboratory expenses (USD 6541).

Among the patients enrolled in HaH, the average duration spent in the hospital was 13.0 days, and 22.6 days at the HaH unit, totaling 35.5 days. The average length of stay for patients in the HaH program was slightly higher than for those treated solely in the hospital (35.5 vs. 33.9 days) (Table 5). The extended hospital stays for HaH patients can be attributed to the reservation of apartments for the patients and their caregivers for a period of 4 weeks. Consequently, there was no need to discharge the patient from HaH before the reservation period concluded.

The average daily ward cost for patients in the INH group was USD 2341. For the HaH group, the ward cost in the hospital was assumed to be similar, while the mean daily ward cost in the HaH unit was calculated to be USD 960. This cost encompassed not only ward expenses such as nursing and physician services but also transportation for healthcare personnel and leasing of the housing. In total, the mean ward cost for the INH group was calculated to be USD 86,396, while the mean total ward cost for the HaH group amounted to USD 52,762, representing a 39% reduction. The total costs for patients in the HaH group were USD 70,896 (not reported in tables), which was 33% lower than the USD 104,530 for the in-hospital group (Table 4). The total cost differences between the two groups were statistically significant at the 0.001% level, based on a *t*-test.

### 3.3. Sensitivity Analyses

In alternative specifications we relaxed the assumption of fixed ward cost in the in-hospital group and assumed that ward cost per day was 10% (alternatively 20%) lower during the follow-up phase than during the conditioning and transplantation phase. This gave daily ward costs of the conditioning and transplantation phase of USD 2595(USD 2779) and daily ward costs in the follow-up phase of USD 2332 (USD 2223). The gains from HaH will be slightly reduced under these assumptions. However, the ward costs are still significantly lower for the HaH patients than for INH patients, 36% lower in the case where the ward costs are assumed to be 10% lower in the follow-up phase that in the conditioning and transplantation phase and 33% lower in the case where the ward costs are assumed to be 20% lower in the follow-up phase that in the conditioning and transplantation phase. In the last alternative, total costs were 28% lower than in the INH patient group.

## 4. Discussion

Allo-HSCT is considered an expensive therapy [18,19], yet it has been found to be cost-effective [20]. In 2022, 19,011 allo-HSCTs were performed in Europe [2] and the total costs were considered very high, prompting a strong demand for efforts to reduce costs without deleterious effects on outcomes. Treating patients at home instead of in the hospital may be an approach to reduce costs in allo-HSCT [18,21]. Since 2019, we have offered patients undergoing allo-HSCT the option of hospital at home (HaH) on a regular basis. The primary aim of this study was to compare the total cost for the immediate peritransplant period for allo-HSCT at home versus allo-HSCT in the hospital.

In this retrospective study, we found that the cost for the peritransplant period could be reduced by up to 33% by treating allo-HSCT recipients at home instead of in the hospital. During the study period, 24% of the allo-HSCT recipients were treated at home, but our data from 2021 and 2022 indicate that at least a third of the patients scheduled for allo-HSCT are candidates for HaH. Svahn et al. [3] reported that 20% of their allo-HSCT recipients were treated at home and that 34% of the allo-HSCT recipients were living within one hour’s driving distance from the hospital. In Norway, where over 90% of allo-HSCT recipients were treated at Oslo University Hospital, less than 30% lived within one hour’s driving distance. Access to four two-bedroom apartments in May 2020 was vital to include 24% of allo-HSCT recipients in our HaH program. It is likely that the COVID-19 pandemic had a positive impact on recruiting patients to the HaH program in 2021 and 2022. The pandemic made working from home common practice, which likely made patients and their caregivers realize the feasibility of participating in HaH while continuing to work. As working from home became less common after the pandemic, it may explain the decrease in the proportion of allo-HSCT recipients treated at home in 2023.

The characteristics of the patients treated at home are comparable to those of all the patients undergoing allo-HSCT at our institution [22]. Svahn et al. [7] found an overrepresentation of men among allo-HSCT recipients treated at home (69%), which aligns with our experience (64% men). However, this is only slightly higher (64% vs.62%), but not significantly higher, than what we reported for all allo-HSCT recipients between 2015 and 2021 [22].

The transplantation team at Huddinge Hospital in Stockholm, Sweden, has shown superior long-term outcomes for patients treated at home compared to those treated in the hospital [8]. Non-relapse mortality was lower and overall survival higher in patients treated at home, while chronic graft-versus-host disease (GvHD) and relapse rates did not differ. This suggests that treatment costs beyond the peritransplant period are likely to be lower for patients treated at home. Like others [3,4,5,6,7,8,9], we have found HaH for allo-HSCT recipients to be safe and advantageous, and the opportunity to stay at home is very well perceived by the patients and their spouses [14].

Healthcare costs in general have increased steadily at a faster rate than inflation, and allo-HSCT has been regarded an expensive therapy [18,19,23]. Therefore, measures to reduce costs associated with allo-HSCT are needed, and HaH has been proposed as a cost-reducing strategy [9]. According to our calculations cost may be reduced by up to 33% when allo-HSCT recipients are offered HaH as compared to being treated in hospital. Ringden et al. [9] found costs to be reduced by 21% when treating children and adolescents at home. Gonzales et al. found in their review of studies on health outcomes, experience of care, and costs that costs could be reduced by 19% with HaH for allo-HSCT recipients [21]. An alternative not analyzed in our setting is more intensive use of outpatient consultations for patients living near the hospital.

The limitations of this study include the following: (i) the cost estimations are from the hospital’s perspective and do not consider societal costs, (ii) it is retrospective in its nature, and (iii) the analysis was conducted at a single center. Although all patients were consecutively recruited, some degree of selection bias may be present. The median age of HaH patients was 54 years, slightly lower but not significantly different from the median age of in-hospital (INH) patients, which was 59 years. This suggests that treatment complications are expected to be higher in INH patients, as age is a significant risk factor in allogeneic stem cell transplantation. However, a significantly higher proportion of HaH patients received myeloablative conditioning, which is associated with higher rates of treatment complications.

Due to data limitations, we were unable to perform matching procedures or regression analyses to address potential selection biases. Consequently, our results should be considered indicative. Additional analyses that explore variability in other cost components than in-hospital care (e.g., housing) would further improve the robustness of the findings.

## 5. Conclusions

Our study indicates that hospital-at-home (HaH) care following allogeneic stem cell transplantation is a safe option. Additionally, our findings reveal that significant cost reductions can be achieved by treating patients in a home setting. This is particularly advantageous for transplant centers, such as ours, where the majority of patients reside more than a 45 min drive away from the hospital. Having access to apartments for these patients is highly beneficial.

## Figures and Tables

**Table 1 healthcare-13-01648-t001:** Number of patients receiving allogeneic hematopoietic cell transplantation (allo-HSCT), 2019–2023.

	2019	2020	2021	2022	2023	Total
allo-HSCT in hospital (INH)	113	97	76	70	87	443
allo-HSCT at home (HaH)	7	30	46	33	27	143
Total	120	127	122	102	114	586
allo-HSCT at home (%)	5.8%	23.6%	37.7%	32.0%	23.7%	24.4%

**Table 2 healthcare-13-01648-t002:** Patient characteristics. N (%) or as described in table. *p*-values from *t*-tests.

Factor	HaH	Hospital	*p*-Value
Number	143	443	
Age (mean, range)	54 (18–74)	59 (15–75)	0.002
Sex (M/F)	86/57	261/182	0.77
Previous HCT %	14	16	0.67
Diagnose:			
Acute Leukemia	83 (58%)	217 (49%)	0.11
Chronic Leukemia	4 (3%)	22 (5%)	0.38
MDS	24 (17%)	85 (19%)	0.57
MPN	3 (2%)	32 (7%)	0.04
MF	7 (5%)	25 (6%)	0.88
Lymphoma	12 (8%)	41 (9%)	0.86
Non-malignancy	10 (7%)	17 (4%)	0.19

**Table 3 healthcare-13-01648-t003:** Staging and donor types.

	HaH	Hospital	*p*-Value
Stage:			
CR 1	51%	44%	
CR 2–3	13%	13%	
CR 4	3%	6%	
Missing	33%	37%	
Donor type:			
MRD	17 (12)	46 (10)	0.75
MUD	102 (71)	299 (68)	0.54
Haplo	3 (2)	30 (7)	0.06
MM URD	21 (15)	64 (15)	0.92
Donor Age (mean, range)	28 (17–60)	28 (16–69)	0.34
Female to Male	16 (11)	27 (6)	0.07
CD34+ cell dose (×10^6^/L)	7.2 (1.2–10.9)	6.9 (0.4–16.2)	0.52
BM/PBSC	15/128	44/395	0.54
MAC/RIC	83/60	186/253	0.005
TBI-based	30 (21%)	44 (10%)	0.001
GVHD prophylaxis:			0.002
CsA + MTX	126 (88%)	335 (76%)	0.004
CsA + MMF	4 (3%)	9 (2%)	0.84
CsA + Sirolimus	1 (1%)	23 (5%)	0.03
PTCy	12 (8%)	68 (15%)	<0.05
Other	0 (0%)	4 (1%)	
ATG	123 (86%)	344 (78%)	0.06
Rec CMV (n/p)	48/95	133/306	0.53
Don CMV (n/p)	62/81	207/232	0.49
Follow-up (year)	2.4 (0.6–5.4)	2.9 (0.5–5.2)	0.02

M: male, F: female, HCT: hematopoietic cell transplantation, MDS: myelodysplastic syndrome, MPN: myeloproliferative neoplasm, MF: myelofibrosis, CR: complete remission, MRD: matched related donor, MUD: matched unrelated donor, HID: haploidentical donor, MM URD: mismatched unrelated donor, Female to Male: female donor to male recipient, BM: bone marrow, PBSC: peripheral-blood stem cells, MAC: myeloablative conditioning, RIC: reduced-intensity conditioning, TBI: total-body irradiation, GVHD: graft-versus-host disease, CsA: cyclosporine, MTX: methotrexate, MMF: mycophenolate mofetil, PTCy: post-transplant cyclophosphamide, ATG: anti-thymocyte globulin, CMV: cytomegalovirus, n: negative, p: positive.

**Table 4 healthcare-13-01648-t004:** Average patient cost per service and in total (USD) for DRG481B (2023 prices).

Type of Service	Cost Per Patient (USD)
Surgery	2074
ICU	6386
Radiology	1349
Laboratory	6541
Radiation	538
Outpatient services	1246
Hospital ward	86,396
Total	104,530

**Table 5 healthcare-13-01648-t005:** Cost of care.

	INH Patient Group N = 443	HaH Patient Group N = 143
Mean length of stay in hospital (days)	33.9 (32.2–36.0)	13.0 (0.7–41.1)
Mean length of stay at HaH unit (days)	0.0	22.6 (1.0–74.4)
Mean total length of stay (days)	33.9 (32.2–36.0)	35.5 (14.2–83.3)
Mean ward cost per day in hospital (USD)	2341.3	2341.3
Mean ward cost per day in HaH (USD)	n.a.	960.0
Mean total ward costs (USD)	86,396.0	54,870.7
Costs before HaH		57%
Costs at HaH		43%
HaH transportation costs in % of total HaH costs		0.06%
HaH Home lease costs in % of total HaH costs		8.79%

## Data Availability

All data are available on request from VKM.

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
