# Peer review of "Hospital at Home Following Allogeneic Hematopoietic Stem Cell Transplantation: An Economic Analysis"

_healthcare, 2025, doi:10.3390/healthcare13141648_

Round 1

Reviewer 1 Report

Comments and Suggestions for Authors

This is a retrospective study where the economic benefits of home care in allogeneic transplant patients are documented. The data are solid and properly presented.

Many transplant centers experience less beds inhouse and more transplant referrals. Home care is an execellent solution, and studies like this with the economic aspect could be useful in argumentation to administrators for organizing the transplant centers activities.

In the discussion, I think it would be noteworthy to mention the "intermediate" solution; That patients could be treated in the outpatient clinic, those living nearby from home, and those with a long driving distance staying at a "hospital hotel". This solution benefits from more efficient use of nurses time.

Author Response

This is a retrospective study where the economic benefits of home care in allogeneic transplant patients are documented. The data are solid and properly presented.

Many transplant centers experience less beds inhouse and more transplant referrals. Home care is an execellent solution, and studies like this with the economic aspect could be useful in argumentation to administrators for organizing the transplant centers activities.

In the discussion, I think it would be noteworthy to mention the "intermediate" solution; That patients could be treated in the outpatient clinic, those living nearby from home, and those with a long driving distance staying at a "hospital hotel". This solution benefits from more efficient use of nurses time.

Response:

We agree with Reviewer 1’s suggestion and have included this as an option in the discussion. (line 253)

Reviewer 2 Report

Comments and Suggestions for Authors

Dear Sir

I have read with attention and interest the contribution: Hospital at Home following Allogenic Hematopoietic Stem Cell Transpalntation: An Economic Analysis. By Vinod Mishra and coworkers

The study presents an interesting comparison of the healthcare costs related to the care of patients after hematopoietic stem cell transplantation comparing a classic "Hospital based" care setting with an innovative "at home" care setting. Surely this original approach is the strength of the study.
Obviously, and this is an intrinsic limit of all cost analyses, the data reported in the study are valid only in the particular Norwegian setting. However, the model presented and the analysis conducted are interesting.Another important limitation of the study is the inability to evaluate the costs related to the commitment of care-gyvers in the extra-hospital setting.

In my opinion the study can be accepted for publication after minor changes.

A first observation on the title: perhaps it would be better to use the expression Stem Cells Transplant rather than Stem Cell Tranplantation.

In my opinion the Authors should carefully review the results section, checking that the data reported in the text are those present in the table.
Page 4, line 148 and following) it is indicated that the patients in home care were 93 males and 51 females (however, the sum is 144 and not 143) while in the table 86 males and 57 females are indicated (the sum in this case is 143).
Page 4, line 152) the statistical significance of the difference in patients treated with myeloablative therapy in the text is indicated as p<0.005 while in the table as p=0.002.

Furthermore, Table II is very long, unclear and difficult to understand and it is recommended to review it, here are some suggestions:
First I would divide Table II into three sub-tables: the first reporting the characteristics of the patients, the second dedicated to the type of donor, and the third dedicated to the prevention of GVHD. In fact, the subdivision is already present but personally I would find it easier to follow and understand three separate tables rather than a single complex table.
Age 54 (18-74) mean, minimum and maximum or median and interquartile range. It would be better to specify.
Previous HCT 20 (14) better to indicate 20 (14%). The same goes for the following rows.
Stage (CR1/CR2-3/No CR) missing: perhaps it would be clearer to prepare four rows:
CR1 73 (51%)
CR2-3 19 (13%)
No CR 4 (3%)
Missing 47 (33%)

Author Response

In my opinion the Authors should carefully review the results section, checking that the data reported in the text are those present in the table.

Page 4, line 148 and following) it is indicated that the patients in home care were 93 males and 51 females (however, the sum is 144 and not 143) while in the table 86 males and 57 females are indicated (the sum in this case is 143).

Among the HaH patients, 86 patients (60%) were male, and 57 patients (40%) were female, ( Line 148)

Page 4, line 152) Changed in txt and tabel<0.005

Response:

The tables and text are now updated as suggested.

Furthermore, Table II is very long, unclear and difficult to understand and it is recommended to review it, here are some suggestions:

First I would divide Table II into three sub-tables: the first reporting the characteristics of the patients, the second dedicated to the type of donor, and the third dedicated to the prevention of GVHD. In fact, the subdivision is already present but personally I would find it easier to follow and understand three separate tables rather than a single complex table.

Age 54 (18-74) mean, minimum and maximum or median and interquartile range. It would be better to specify.

Previous HCT 20 (14) better to indicate 20 (14%). The same goes for the following rows.

Stage (CR1/CR2-3/No CR) missing: perhaps it would be clearer to prepare four rows:

CR1 73 (51%)

CR2-3 19 (13%)

No CR 4 (3%)

Missing 47 (33%)

Response:

We have updated to expressions to reflect the suggestions. Prepared four rows for staging information

Tabel one contains information characteristics of the patients. However, after careful consideration, we decided to keep new Tabel two contains information of donors and GVHD. In our opinion, the current table structure provides the best representation of the data..

Reviewer 3 Report

Comments and Suggestions for Authors

The study evaluating the cost of delivering allogeneic HSCT via a hospital-at-home model compared to in-hospital care is timely, given the increasing healthcare burden and the potential to improve healthcare resource allocation. The study provides valuable insight into cost difference. However, I think there are several questions need further clarification to strengthen the validity and interpretability of the findings.

  1. While the study focuses on cost comparison between 2 groups, I would expect a clearer contextualization of these results within the broader scope of patient outcomes. Were there any observed differences in readmissions, complications, or survival that could help readers interpret whether lower costs in the HaH group also reflect equivalent or better outcomes? Referencing prior studies on clinical safety and effectiveness of HaH in the background can strengthen the rationale for the cost comparison. Without knowing the a bit more details on outcomes, the cost reduction could be misleading.
  2. It's unclear to me how the control (in-hospital) group was selected. Were patients matched on clinical or demographic variables? More importantly,  it's also not clear to me whether any statistical adjustment was made to account for baseline differences between groups, such as age or comorbidities, etc. This raises concerns about potential confounding and selection bias, which should be explicitly addressed and, if possible, corrected through adjusted regression analyses, PSM, etc.
  3. The cost analysis lacks a clear statistical analysis plan. There is no mention of formal hypothesis testing, confidence intervals, or regression analysis to determine whether observed cost differences are statistically significant or may be due to chance. The absence of uncertainty measures limits confidence in the findings. I would suggest that authors should consider including appropriate statistical testing and clarifying whether cost comparisons were adjusted for patient characteristics. 
  4. Lastly, the cost estimate for the HaH group may be influenced by factors such as the 4-week fixed apartment reservation, which could overstate actual HaH costs. Also, I noticed that the sensitivity analysis only addresses variation in in-hospital ward costs. Additional analyses that explore variability in other cost components (e.g., housing) would improve the robustness of the findings. 

Author Response

Reviewer 3

The study evaluating the cost of delivering allogeneic HSCT via a hospital-at-home model compared to in-hospital care is timely, given the increasing healthcare burden and the potential to improve healthcare resource allocation. The study provides valuable insight into cost difference. However, I think there are several questions need further clarification to strengthen the validity and interpretability of the findings.

  1. While the study focuses on cost comparison between 2 groups, I would expect a clearer contextualization of these results within the broader scope of patient outcomes. Were there any observed differences in readmissions, complications, or survival that could help readers interpret whether lower costs in the HaH group also reflect equivalent or better outcomes? Referencing prior studies on clinical safety and effectiveness of HaH in the background can strengthen the rationale for the cost comparison. Without knowing the a bit more details on outcomes, the cost reduction could be misleading.

Response:

We concur with the initial points raised by Reviewer 3. The article already references several studies at the beginning that describe the effects of Hospital at Home (HaH) on patient safety and other outcomes. ( Line-300-304)

  1. It's unclear to me how the control (in-hospital) group was selected. Were patients matched on clinical or demographic variables? More importantly, it's also not clear to me whether any statistical adjustment was made to account for baseline differences between groups, such as age or comorbidities, etc. This raises concerns about potential confounding and selection bias, which should be explicitly addressed and, if possible, corrected through adjusted regression analyses, PSM, etc.

Response:

Although the patients were consecutively recruited, we agree that selection is a potential weakness in our study and have now discussed this more in details. ( Line 254-260)

  1. The cost analysis lacks a clear statistical analysis plan. There is no mention of formal hypothesis testing, confidence intervals, or regression analysis to determine whether observed cost differences are statistically significant or may be due to chance. The absence of uncertainty measures limits confidence in the findings. I would suggest that authors should consider including appropriate statistical testing and clarifying whether cost comparisons were adjusted for patient characteristics. 

Response:

We agree with the reviewer’s comments and have included a paragraph describing the significance level based on a t-test. However, due to limitation in our data material, regression analyses were not performed. This is now stated in the limitation section. Line 267-270

Lastly, the cost estimate for the HaH group may be influenced by factors such as the 4-week fixed apartment reservation, which could overstate actual HaH costs. Also, I noticed that the sensitivity analysis only addresses variation in in-hospital ward costs. Additional analyses that explore variability in other cost components (e.g., housing) would improve the robustness of the findings. 

Response:

We have added this in the limitation section. Line 268-270

Round 2

Reviewer 3 Report

Comments and Suggestions for Authors

The revised version looks good to me